# Smart Justice in Italy: Cases of Apps Created by Lawyers for Lawyers and Beyond

**Giampiero Lupo [1,\*] and Davide Carnevali [2,\*]**

1   Institute of Applied Sciences and Intelligent Systems, National Research Council of Italy (ISASI-CNR), 73100 Lecce, Italy
2   Institute of Legal Informatics and Judicial Systems, National Research Council of Italy (IGSG-CNR), 40126 Bologna, Italy
\*   Correspondence: giampiero.lupo@cnr.it (G.L.); davide.carnevali@cnr.it (D.C.)

**Abstract:** The smart city literature states that three levels of institutional layers (regulatory, normative, and cognitive) and four typologies of actors (government, universities, citizens, and the private sector) support private initiative for developing smart technologies. Focusing on the emergent phenomenon of smart apps ideated by lawyers' private initiatives, this paper acknowledges that other factors, including the ubiquity of mobile technologies and the absence of effective public services provided by public institutions, contribute to the institutional and organizational humus necessary for the creation of intelligent technological proposals. In the light of the organizational theory framework, and based on the analysis of the literature on smart cities and e-justice and on the empirical investigation of two Italian lawyers' apps (Collega and Anthea), this paper identifies the institutional, organizational, and technological conditions under which smart technologies are being developed in high-regulated public institutions' contexts as justice systems. The findings of the study described in this paper help integrate the contribution of the literature on the topic.

**Keywords:** smart city; smart technologies; e-justice; ICT for lawyers; justice administration

## 1. Introduction

The diffusion of emerging technologies, such as the Internet of Things (IoT) and mobile technologies, contributed to the evolution of the high-tech innovation economy that observes more and more private unknown inventors as principal actors than industry colossuses. The examples of Summly created by 17-year-old Nicholas D'Aloisio and purchased by Yahoo for 30 million dollars, or Napster, founded by 18-year-old Shawn Fanning, acknowledge this phenomenon. Thanks to the initiatives of individual entrepreneurs, an indecipherable number of apps that make life easier for users have been created. For example, with a few taps on the touchscreen, we can book a "car sharing" car, arrange a bank transfer, or pay the train ticket. The phenomenon of the creation of apps and online services by individuals also spreads to the legal profession with the diffusion of apps and online technologies that are ideated by lawyers for justice professionals and citizens (Hamm 2013; Gershowitz 2019). The development of legal technology conceived by lawyers enriches the justice environment with more and more diffusion of increasingly complex technology, such as artificial intelligence systems, smart contracts, digital identity, and blockchain systems for supporting justice professionals (Coelho et al. 2019).

Thanks to the availability and ubiquity of technological modules, such as mobile phones and global positioning systems (GPS), it is possible for a nonexpert user (as a lawyer with a limited expertise in information and communication technologies (ICT)) to ideate apps and IT innovation by combining different modules and by exploiting the eventual availability of funds and ICT companies. These technologies are created in competition with technologies launched by public administration and software houses. However, in some cases presented later in this article (Section 4), ICTs created by the private initiatives

of lawyers for justice professionals bridge the gap between public administration services incapable of keeping pace with technological evolution. Like other areas of application, these forms of innovation can be defined as subject-matter expert-driven IT developments (Prior et al. 2011; Potts et al. 2010) (different from simple private-driven innovations) due to the requirement of legal expertise for proposing expert technological solutions.

This article explores the conditions under which private initiative designs, develops, and commercializes online and app technologies for providing services to justice professionals and citizens. With an organizational theory approach, the present authors explored the previous literature analyses on the topic—smart city and e-justice—to frame an interpretative framework useful for empirically analyzing two example apps (and relative online services) that have been recently developed for lawyers: Collega, an Italian app for searching and engaging a substitute for a lawyer in a hearing, and Anthea, an app for supporting communication and document exchange among the various actors involved in divorce cases. The two case studies were selected because they represent clear examples of technology—ubiquitous, IoT, and modular—devised by a nonexpert user in a hostile institutional context. By focusing only on the two case studies, which were selected among numerous commercial legal service apps developed by law firms, it is possible to conduct an in-depth and qualitative analysis to highlight the most crucial conditions for technological development, particularly in high-regulated public institution contexts such as justice systems. Furthermore, selecting the case studies in the Italian legal context gave us easier access to data and facilitated the operationalization of the in-depth analysis. In addition to this, we analyzed a legal context consistent with other civil and even common law countries in the sphere of influence of international organizations driving the harmonization of law such as the European Union, the Council of Europe, and the United Nations.

The literature review in Section 2 introduces the smart cities' contributions (Albino et al. 2015; Kourtit et al. 2013; Leydesdorff and Deakin 2011) useful for clarifying the conditions under which smart technological solutions are implemented for supporting public services and improving public value. These conditions primarily refer to the role of institutional settings (regulatory, normative, and cognitive institutions) and different typologies of actors (government, universities, citizens, and private sector). However, to interpret the emergent phenomenon of smart apps developed by lawyers' private initiative, it is necessary to go beyond the smart city explanation and introduce other concepts. These concepts refer to the technological characteristics of the developed apps as the ubiquity of mobile technologies, IoT, and their modularity (Hanseth and Lyytinen 2016; Contini and Lanzara 2009; Lupo 2014). In addition, in the case of apps and online technologies for lawyers, as we will see later in this article (Section 5), the absence of an existing public service creates the institutional humus useful for designing and developing intelligent technological proposals. The analysis of the two case studies in Section 5 acknowledges that all the mentioned factors contribute to the creation of useful apps for justice professionals and citizens by ICT nonexperts such as lawyers. The empirical analysis and the investigation of theoretical concepts, mainly deriving from the organizational theory, help to clarify the phenomenon related to e-justice system development.

This article is organized as follows. First, we introduce the theoretical framework based on the literature review on the topic (Section 2). Second, we describe the methodology of research (Section 3). Third, Section 4 briefly introduces the two apps selected for the analysis. Fourth, we apply the theoretical framework to interpret the data on the two analyzed systems with the objective of drawing conclusions on the factors and conditions supporting technological design and development (Section 5). Finally, a conclusive session wraps up the results of the analysis.

## 2. Theoretical Framework

The scope of this study is to acknowledge that in particular conditions actors with an elementary ICT knowledge can primarily contribute to the ideation of smart technologies. The characteristics of modern ICT technologies, including mobile phones as integrated

tools or IoT, favor this phenomenon. These technologies are ubiquitous, easily available, and modular. Therefore, a "smart" idea, even if not supported by a technical ICT expertise, allows development of an application around a certain task.

The literature on smart city and smart technology (Kummitha and Crutzen 2017; Kitchin 2014; Manolova et al. 2008) explored the conditions under which the private initiative is the main lever creating the smart technologies. Initially, the smart city reflective school of thought confirmed the interconnection between the implementation and diffusion of technologies and human capital. The introduction of technologies, such as smartphones, requires effort from users in terms of acquiring new technological skills to use their applications (Leorke et al. 2018). Simultaneously, the continuous use of the technology allows gaining new skills to broaden user competence and reduce the "threshold between the maximum level of feasible simplicity versus the maximum level of manageable complexity" (Contini and Lanzara 2014).[1]

The rationalistic or pragmatic school of thought (Orlikowski 2000; Toppeta 2010; Neirotti et al. 2014) criticized this approach by reaffirming the importance of private initiative in the design and development of smart technologies. For example, Orlikowski (Orlikowski 1992, 2000) switched the direction of the relationship between technology and human capital and affirmed that often the knowledge of communities and users is the basis of the introduction of new technologies. Private initiative and a high level of human capital are the primary levers for the diffusion of new ideas impacting technological development and diffusion (Giffinger and Pichler-Milanović 2007; Toppeta 2010). Therefore, in addition to focusing on technological development, smart cities should give significant attention to enabling citizens to enhance their capabilities and human capital. This makes citizens more aware of their needs and issues and allows them to utilize their skills and capabilities to invent and promote the usage of technology (Orlikowski 2000). This perspective has opened space for debates regarding the roles of various stakeholders, which could lead to ideal planning and execution of smart technologies (Bunnell 2015). As per the smart city literature, the innovation arises from the involvement of stakeholders coming from different contexts and experiences. In order to be smart, cities should significantly emphasize innovative partnerships. Therefore, various sectors and stakeholders must come together to promote entrepreneurship and innovation among citizens (Giffinger and Pichler-Milanović 2007). In addition, the literature on e-justice highlights the involvement of stakeholders and users in the design and implementation of technology by introducing the "psychological and political/power aspects" (Mohr and Contini 2011) of a technological change (Agrawal and Vieira 2013; Andrade and Joia 2012; Lupo and Bailey 2014). Based on this concept, Lupo and Bailey (Lupo and Bailey 2014) referred to the advantages of an iterative process that incorporates feedbacks from key stakeholders, thus fostering the stakeholders' acceptance and ownership of technological change (Bailey et al. 2013a; Agrawal and Vieira 2013).

Beginning from the concept of stakeholder involvement and partnership, the smart city literature acknowledges that innovations are quite often inspired by a "triple-helix model" in which universities, industries, and governments engage with each other to create a productive infrastructure for promoting bottom-up interventions (Leydesdorff and Deakin 2011). A supportive environment with guaranteed channels of communications, resources, and technological infrastructures favors the creation and proliferation of this ecosystem based on the connection of the three pillars (university, industry, and government). Successively, the smart city literature (Calzada and Cobo 2015; Kummitha and Crutzen 2017) further articulated this concept by including a fourth pillar, the citizens. The "quadruple-helix model" traces the ideal path for innovation by focusing on the coop-

---

[1] Lanzara (2004) stresses that successful ICT systems have to achieve the right balance between a system's maximum level of feasible simplicity and its maximum level of manageable complexity. As Lanzara notes, systems that are simplified to a point that undermines the functionalities, value and usefulness are highly unlikely to attract users, and may in fact drive users to offline procedures (Lanzara 2004). On the other hand, systems cannot be so complex as to be beyond the technological capacity of most users. Designers, Lanzara argues, should take into account the two thresholds, and implement strategies for keeping systems in the space between the maximum manageable complexity and minimum feasible simplicity.

eration among the private sector, government, citizens, and universities. Nam and Pardo (Nam and Pardo 2011) stated that it is almost impossible to imagine an inclusive smart city where public institutions, private sector, voluntary sector, and citizens do not cooperate with each other in innovative projects.

The inclusive approach of the smart city literature fails to consider other environmental conditions that may positively affect technological developments and the creation of services based on ICT and mobile technology, above all in high-regulated public institution contexts such as justice. For instance, the case of a smart app for lawyers (discussed herein) acknowledges that the lack of a public service supporting lawyers' interaction with tribunals pushes the lawyers to fill the "public services" void by designing and contributing to the implementation of smart technologies. The lack of public services associated to a ubiquitous and easy to access technology may foster the participation and involvement of citizens in smart city innovations.

Lanzara (1993) described the relationship between the lack of public services and private entrepreneurship by focusing on the concept of "negative capabilities". He introduced the concept of negative capability in his 1993 book that analyzes the conditions under which a human innovates and creates values in critical and hostile environments (Lanzara 1993). The concept "negative capability" derives from a quote from the English poet John Keats who defined negative capabilities as "when man is capable of being in uncertainties, mysteries, doubts, without any irritable reaching after fact and reason" (Keats 1817; Lanzara 1993). Notably, Lanzara's analysis on services created by private citizens is based on a small mobile coffee bar and a logistic system created for gathering and distributing rescue equipment in the aftermath of the 1980 great earthquake in Irpinia, Italy. For Lanzara, the negative capability, which is the ability of private citizens to adapt and create useful practices, services, and routines in a hostile environment, arises from the void of usual frameworks, infrastructures, and services provided by government and public administration. Moreover, a private initiative compared with a government action is more free and adaptive given that it does not (or does to a lesser extent) have to comply to the mechanisms and rules of complex bureaucracy (Ciborra and Lanzara 2017). Lanzara's arguments provided an adequate framework for describing the private initiative of lawyers having an elementary ICT knowledge in the designing and contribution to implement smart technologies for their colleagues. The absence of a public service supporting lawyers' interaction with courts represents fertile humus for the development of the negative capabilities that are useful for implementing technological innovations.

Lanzara's theses and the inclusive approach of the smart city literature provide the theoretical basis for the empirical analysis of the two example lawyers' apps, namely Collega and Anthea, selected for this study.

## 3. Methodology

Based on qualitative research methods and techniques, the empirical analysis presented here utilizes an interdisciplinary approach combining social science, legal, and ICT theories (Clarke 2014). This methodology is adequate for investigating technologies developed in the justice context, which are cross domain by nature (involving public, institutional, or legal organizational contexts). Moreover, the analysis is inherently qualitative as it excludes the quantitative analysis of a large sample of cases and focuses more in depth on a few cases (case-study approach) (Yin 1987). Based on this, the investigation focuses on the two Italian apps selected between numerous examples of technologies enabled by law firms to investigate their characteristics of being designed by ICT inexperienced individuals who are also potential users of the system. As anticipated, the case studies selected concern the Italian legal context. This selection guaranteed to investigate in-depth jurisdiction consistent with other civil and even common law countries because it has been, and still is, influenced in terms of law evolution by all international organizations (the European Union, the Council of Europe, and the United Nations) in which it is involved. In addition,

the cases investigated (children management in divorce cases and lawyer substitution in proceedings) are common legal subjects in almost all national juridical contexts.

The empirical analysis is based on the case-study approach because this method is shown to be the most effective way to study ICT phenomena in the broad area of justice (Fabri and Langbroek 2000; Contini and Fabri 2003; i Martínez and Fabra i Abat 2009; Contini and Lanzara 2014; Rosa et al. 2013; Velicogna 2007). In-depth case studies are the preferred strategy when dealing with "how", "who", or "in which way" questions. The researcher/author has little control over events and the focus is on a contemporary phenomenon within some real-life context (Yin 2003). Furthermore, the in-depth case study methodology allows the use of an interdisciplinary approach, which is particularly relevant in an area where multiple factors (such as legal, institutional, organizational, technological, and practical) are deeply intertwined (i Martínez and Fabra i Abat 2009; Contini and Lanzara 2014; Contini and Fabri 2003; Velicogna et al. 2011).

In particular, data regarding the study of the two apps have been collected through a qualitative analysis of the relevant documentation developed by the two projects (website, official description of the app, and previous interviews of developers on the media) with a focus on the functioning, the infrastructure components (organizational, technological, and legal), and the history of development. The analysis of documentation is associated with the administration and analysis of semi-structured interviews (McIntosh and Morse 2015; Horton et al. 2004; Schmidt 2004; Smith 1995) to relevant actors (the founders of the apps).[2] The interviews, which lasted 60 min on average, covered several topics regarding the generic functioning of the system, the history of development, and the relationship with public administration (the topics of the semi-structured interviews are listed in Table 1). The narrative analysis (Webster and Mertova 2007) of data collected through the documentation and semi-structured interviews of in-depth case studies has been cross referenced to assess the COLLEGA and ANTHEA experience in terms of adherence to the theoretical principles associated with the smart city and e-justice literature described in the next section.

**Table 1.** Topics of Interviews.

| | | | |
|---|---|---|---|
| 1. | Role, training, and professional experience of the interviewee | 9. | User involvement |
| 2. | ICT literacy | 10. | Test policy |
| 3. | History of app development | 11. | System components |
| 4. | Barriers in development | 12. | User identification |
| 5. | Competences activated for development | 13. | E-payment |
| 6. | Relationship with public administration for app development | 14. | Security of systems |
| 7. | Other institutions and actors involved | 15. | Diffusion of system among users and effects on user work routines |
| 8. | Installed base components | 16. | Revenues from investments |

**Note:** List of topics of the semi-structured interviews for empirical analysis data gathering.

## 4. Investigated Apps (A Brief Description)

*4.1. An App for Lawyers' Substitutions: COLLEGA*

COLLEGA (in English *colleague*)[3] is an app born from the idea of an Italian lawyer. It can be used to find a domiciliary, a substitute for a hearing, or a colleague who can

---

[2] The Semi-structured interview involving the following arguments: role of interviewed; system description; system development; actors involved in development; test; system infrastructure; relationship with institutions; diffusion of the service. Average duration of the interview: 60 min.

[3] The information provided in this section derives from the COLLEGA Official website (www.collegaonline.it; accessed on 30 June 2021) and from the semi-structured interview with the creator of the app.

carry out any type of administrative activity in one of the Italian judicial offices.[4] This app smartly combines several technologies and functionalities, such as geolocalization, internal chat, and exchange of documents, to provide an online service that facilitates the routine operations of a regular lawyer.

For all lawyers registered to the Italian Bar Association[5] can use this app. To access the app, users need to register by providing personal data, particularly, a mobile phone number and a valid email address. The app sends a 4-digit code to the mobile number for allowing the access.

Once registered, users can access the app's functionalities through a set of dedicated screens: 1. Search; 2. Favorites; 3. Recent Activities; 4. Board; 5. Menu; 6. Legal Agenda. To be able to pursue a search, a lawyer (or practitioner) needs to complete the profile with all the personal data and information regarding the law firm, such as tax code and VAT number. These data are also necessary because the application allows receiving invoices in case of in-app purchases or invoices coming from colleagues. Once the profile is complete, the lawyer can activate the search function. Further, to look for a substitute, the user needs to indicate the court, the activities to be performed, the time and day of the activity, and the compensation offered (user can indicate a fee or select "to be agreed"). Other confidential information can be shared with the substitute through a dedicated section. The search can be geolocalized so that COLLEGA will exclusively search lawyers who are present in the selected office at that moment and have activated the GPS functionality in the application. The geolocalization is one of the most interesting solutions applied by the application. Moreover, to develop the lawyer localization functions, the COLLEGA designers mapped most of the Italian judicial offices in the Italian territory at all levels (first instance courts, the court of appeal, administrative courts, tax commissions, juvenile courts, and the court of cassation).

The system notifies the lawyers available and present in the selected office where a search has been activated. The lawyer looking for a substitution can only select colleagues who, after reading the search, declare to be available to perform the service. Once a contact is established with a colleague, the system provides his/her telephone number. The user can also communicate with a colleague through an internal chat. Furthermore, this app also allows the filing and sending of any document to the colleague who accepts the assignment. The app keeps track of the tasks awarded, collaborations accepted, and colleagues with whom a lawyer collaborated. These functions also help to create a "Favorite" contacts group to circumscribe the search for a substitute only in the preferred group. Within the favorite group, a lawyer can also include other users directly via the phone book. The app also takes advantage of an agreement with the Associazione Italiana Giovani Avvocati—AIGA (in English *Italian Association of Young Lawyers*). Thanks to the convention, users who are part of the AIGA benefit from the section reserved for them in the application and register via the link provided in the email sent by the association. Additionally, searches can be restricted to AIGA members.

The registration to the service and the first four tasks assigned are free. From the fifth task awarded, it is necessary to pay a yearly subscription (EUR 11.99), which allows receiving and assigning an unlimited number of tasks for the entire duration of the subscription.[6] The fee can be paid directly through the app and with a regular credit card. At the moment, it is not possible to pay the professional services of colleagues directly through the app. The subscription fee allows keeping the app without advertisement, even though previous versions of the app and related website used to show banners and adverts.

---

4    The article 102 of the Code of Criminal Procedure and the law 31 December 2012, n.247 establishes that lawyers can be replaced by another lawyer, with a verbal assignment, or by a qualified practitioner, with written authorization.

5    In Italy to become a lawyer and be registered in the bar association, it is necessary to carry out an internship of at least 18 month and a final exam. Lawyers enroll in the local bar association situated in the district of the court where the lawyer resides (Law 31 December 2012, n.247).

6    The app does not deduct the four free tasks if they are granted to favorites.

A further useful functionality of the app is the bulletin board of the courts and bar associations in which users can publish, after approval, short announcements such as offer/search for rentals, colleagues' services, collaborations, etc. Recently, developers added two more functions to the app: the legal agenda and the management of hearing report. The legal agenda allows users to include events related to appointments, fulfillment, hearings, and the most common details of the cases such as party, counterparty, judicial office, and judge. The management of the hearing report allows creating minutes of the hearing directly from the mobile as well as from the web application and sharing the report privately with the counterparty and judge.[7]

### 4.2. An App for Divorce and Parental Conflict Management: ANTHEA

ANTHEA[8] is an app developed in the context of the ANTHEA Project that provides tools to divorced couples for managing the affairs related to the new family status and children's handling in cases of parental conflicts.[9] In addition, the ANTHEA Project provides a protocol binding the parties in a divorce that involves the use of the app ANTHEA to be approved before the judge in charge of the case.

ANTHEA provides a specific chat to divorced couples with restricted and regulated access that checks that communications are based on a civil and appropriate language. Furthermore, the app allows defining a calendar of shared events to manage payments and reimbursements, to share and comment familiar experiences (e.g., holiday pictures) even to grandparents in protected mode, and to create a shared archive of various documents. This app allows the divorced couples to book an appointment or request a specific request to the social service delegate, as well as consult the judge in charge of the divorce that can decide in real time.

All data produced during the use of the app are collected in different archives that can be used for the proceedings by the social service delegate and the judge in charge. In other words, the app provides a real help to the couple for effectively managing day-by-day divorce interactions. Moreover, it provides to the social service delegate and to the judge in charge a sort of "console" for monitoring the compliance of divorce dispositions.

It is clear that all data provided by ANTHEA are available only if prior and explicit authorization is given by the divorced couple with the acceptance to participate in the ANTHEA Project. At the time of registration and subscription of the license, users accept the methods of use and the related consequences and demonstrate to have read and accepted the system's disclaimers.

Once the application is freely downloaded from the ANTHEA Project's official website (the app is available for iOS and Android operating systems), users deal with the registration to the service. Users have to provide all personal data, bank account details, and their digital references for the connection, authorization, and the references of the social service delegates and the judge in charge. Then, the couple has to indicate the legal agreement's details of the divorce sentence, particularly with reference to the provisions for the custody of the children. Once registered, users pay the license cost (EUR 50 per year) for a single divorced couple (including their parents) and approve the app's legal conditions. Successively, the system sends differentiated access codes for each person involved.

Users have access to the app's functionalities through a set of dedicated modules presented in a menu on the starting page of the app: 1. Communication; 2. Creation, management, and sharing of specific events; 3. Payments and reimbursements; 4. Sharing

---

7 The counterparty and the judge have access to the shared report also outside COLLEGA through a PIN number communicated by the lawyer (www.collegaonline.it; accessed on 30 June 2021).

8 The information provided in this section derives from the ANTHEA Official website (www.progetto-anthea.com; accessed on 30 June 2021) and the semi-structured interview with the creator of the app.

9 In Italy, the Law n. 54 of 8 February 2006 disciplines the cases of minors' maintenance in divorced couples. The fundamental principle is that, even in the event of the divorce of the parents, the minor child has the right to maintain a balanced and ongoing relationship with each of them, to receive care, education and instruction from both and to maintain meaningful relationships with the ancestors and with the relatives of each parental branch.

of familiar experiences (pictures and comments); 5. Create a shared archive of documents; and 6. Archive access for the social services delegate and the judge in charge.

The first function "Communication" is a module that consists of a specific chat interface for the divorced couple moderated by an automatic system that signals the presence of inappropriate language. Chats remain registered in the system for the purposes of evaluation by the social service delegates and the judge in charge. The informal system of a chat (protected and supervised), different from a phone call or indirect communication based on email, is appreciated for generating an appropriate and fruitful climate between the couple.

The second function provides the "Creation, management, and sharing of specific events". The creation of an event allows the couple to engage with each other on a specific topic. For example, assume the case of a request to pick up a child from school; the other parent can accept, reject, or ask for clarification with a dedicated chat. In case of acceptance, the requesting parent can send the geolocation of the event to help the other parent reach the place. This function helps manage variations in the plans established by divorce parties, limiting adverse reactions, miscommunications, and the lack of care toward children.

The "Payments and reimbursements" function operates with the same logic of event management. A member of the couple can ask for the payment or reimbursement of any expense. The app allows paying these expenses by credit or debit card with an automatic receipt. The possibility to provide the payments directly via the app, as well as record and archive the successful payments, creates more convenience for the couple and fewer opportunities for controversy in the management of payments.

"Sharing familiar experiences" is a fourth function that allows the users to share pictures and comments related to experiences involving all the members of the family. In particular, grandparents are often keen to participate in the experiences of their grandchildren. However, it is difficult for the relatives of a divorced couple to access information or testimonies of the daily life of their grandchildren, given that often they cannot participate in all events. The system notifies all involved subjects in case of a user activity on events and allows any user to make a comment on them.

With the fifth function, both parents can "Create a shared archive of documents" to store all documentation related to the management of the child's life. Both parents can collect data for the common interest in a shared archive. The shared archive can be organized in sub-archives related to certain peculiarities, for example, separate folders for school documents, doctor certificates, etc. It is possible to import the document from the smartphone database or take its direct photograph using the phone camera or import it into PDF.

All the mentioned functions allow providing evaluation feedback and comments. At any time, the divorced couple can ask for a consultancy, a direct aid from a social service delegate, or an evaluation from the judge in charge. The overall interactions are recorded in an "Archive access module" (sixth function). Interactions can be easily retrieved through a search engine freely or by using a series of predefined filters. Additionally, the social service delegate can use digital archives to draft reports based on certified data provided by the system. In addition, the judge in charge can react promptly in the case of a specific request using the data collected in the system.

## 5. The Analysis of the Two Systems on the Basis of the Theoretical Framework

This section presents the theoretical implications concluded from the analysis of ANTHEA and COLLEGA. The study of the two apps acknowledges that, in particular conditions, actors with an elementary ICT knowledge can contribute to the development of successful smart technologies for justice professionals. In particular, the analysis allowed the verification of the conditions under which smart technology for lawyers can be developed through the private initiative of a nonexpert user. By studying the two apps, we confirmed the tenets of the previous literature on the topic (see Section 2). In addition, we introduced further implications that regard, for instance, the role of "politics" and the user

base and the importance of an incremental development of systems. We used a deductive method based on the application of previous theories that provide examples of similar phenomena and "stimulate our thinking about properties or dimensions that we can then use to examine the data in front of us" (Strauss and Corbin 1998). In addition, the analysis of the case study made it possible to inductively introduce new theoretical concepts useful for explaining the studied phenomenon.[10] The theoretical framework's elements applied to the analysis of case studies and the theoretical concepts derived inductively from the empirical investigation are summarized in Table 2.

**Table 2.** Theoretical framework factors and empirical analysis concepts.

| Theoretical Framework Factors of Analysis | | | |
|---|---|---|---|
| *Factor* | *Definition* | *The Literature* | *Source* |
| Private initiative | Role of private initiative for innovation | The smart city literature and rationalist school | (Orlikowski 1992; Toppeta 2010; Neirotti et al. 2014; Giffinger et al. 2007) |
| Stakeholders' role | Involvement of stakeholders in technological development | The e-justice and smart city literature | (Mohr and Contini 2011; Agrawal and Vieira 2013; Andrade and Joia 2012; Lupo 2014; Bailey et al. 2013a) |
| Triple-helix model | Role of industries, universities, and governments in technological innovation | The smart city literature | (Leydesdorff and Deakin 2011; Kummitha and Crutzen 2017; Calzada and Cobo 2015) |
| Quadruple-helix model | Role of industries, universities, governments, and citizens in technological innovation | The smart city literature | (Nam and Pardo 2011) |
| Negative capabilities | Capacity of private citizens to adapt and create useful practices, services, and routines in hostile environments | The ICT literature | (Ciborra and Lanzara 2017; Lanzara 1993) |
| Analytical concepts derived from empirical analysis | | | |
| *Factor* | *Definition* | *The Literature* | *Source* |
| Test policy | Reiterated tests involving users for stakeholders' inclusion and acceptance of technology | The ICT and e-justice literature | (Bailey et al. 2013a) |
| Self-reinforcing process | "Critical mass" of users as a significant factor in ICT growth | The ICT and e-justice literature | (Hanseth and Aanestad 2003) |
| Maximum manageable complexity | Entanglements, investments, and maintenance costs considered tolerable in terms of affordability and sustainability | The ICT and e-justice literature | (Carnevali 2019) |
| Incremental approach | Development through iterative process that incorporates feedback from key stakeholders | The ICT and e-justice literature | (Ciborra and Lanzara 2017; Lanzara 1993) |
| Ubiquity of technologies | Human–technology interactions in which the user operates computing systems and equipment simultaneously without being aware that machines are performing tasks | The ICT literature | (Nucera et al. 2018) |
| Installed base | Technological solutions, institutional arrangements, organizational practices, and legal frameworks that are already established when a new system is developed | The ICT and e-justice literature | (Hanseth and Aanestad 2003; Hanseth and Lundberg 2001; Ciborra and Lanzara 1994; Kallinikos 2009; Lanzara 2009; Lupo 2014; Velicogna and Contini 2009) |
| Modularity | System development based on an infrastructure composed of different technological components | The ICT and e-justice literature | (Hanseth and Lyytinen 2016; Lanzara 2009; Lupo 2014) |
| Assemblage structure | Integrated and different loosely coupled layers—organizational, technical, institutional, and regulative—connected to each other | The ICT and e-justice literature | (Cooper 1998; Lanzara 2009) |

**Note:** List of analytical concepts deriving from the theoretical framework and analysis.

The first element that is introduced regards the private initiative of citizens for developing the technology.

---

[10] This combination of inductive and deductive methodology is at the basis of the grounded theory method of analysis. An in dept analysis of grounded theory concepts is out of the scope of the paper. However, it is worth mentioning that if on the one hand grounded theory provides the dictum that "there is a need not to review any of the literature in the substantive area under study" (Glaser 1992, p. 31) for fear of contaminating, constraining, inhibiting, stifling, or impeding the researcher's analysis, on the other hand Strauss and Corbin supported an analysis based on previous theories given that the literature is able to provide examples of similar phenomena (ST).

*5.1. Protagonists of Innovation: The Quadruple-Helix Model, the Private Initiative, and the Role of Users' Base*

The two experiences of smart app development for lawyers emphasize the role of private citizens for technological innovation already evidenced in the previous literature. The rationalistic or pragmatic school of thought of the smart city approach (Neirotti et al. 2014; Orlikowski 1992) affirmed the importance of private initiative in designing and developing smart technologies together with the necessary human capital and knowledge of communities and users. Additionally, the quadruple-helix model developed by scholars of the smart city literature (Calzada and Cobo 2015; Kummitha and Crutzen 2017) states that the technological development results from the cooperation between institutional actors such as the private sector, government, universities, and citizens, thus including users and communities in the ideal path for innovation.

The story of the development of the COLLEGA app represents a characteristic example of the realization of a technological idea by a user not expert in typical ICT skills. The creator of COLLEGA is a lawyer with an amateur knowledge of ICT and no computer programming skills. Despite this, the availability and ubiquity of technological modules, such as mobile phone technology and GPS, allowed the COLLEGA's creator to develop the app.

Furthermore, ANTHEA was created by a technology-loving lawyer without specific computer education and who takes communication as a fundamental issue in court business. Regularly dealing with family and minors' cases, ANTHEA's creator felt the importance of interpersonal relationship attention in handling conflicts in divorce cases, particularly when children are involved, and applied this knowledge in developing the app.

In both cases, the project creators started from an idea based on the combination of ubiquitous technological modules available at the time of the apps' development. The apps' creators had only a basic knowledge of ICT; therefore, in both cases it was necessary to involve an external company to move from the idea to the concrete development. In the COLLEGA case, the involved ICT firm developed the app based on the combination of technologies available in cellular phones (such as microphone, geolocalization, and internet connection) and personal computers connected online. In the ANTHEA case, the creator (the lawyer) and his partner involved a small software house through personal financing for the app's development. The development benefited from the creator's considerable experience in the divorce field. The involvement of the ICT firm reflects the quadruple-helix model (Leydesdorff and Deakin 2011) when it supports the cooperation of citizens with institutional actors, such as private companies.

Users' private initiative for technological development may result in a further positive result. Both cases highlight that the implementation of an app by one of its main users is advantageous as developers can directly test the apps in the first place. This can be defined as a "homemade" bottom-up approach for testing, which is characterized by a creator–developer who acts as a user and directly tests the system on the field. Feedback-based testing and improvements may also involve a larger user base along with the developer's effort. In both cases, in addition to usual reiterated tests, the app developers took advantage of reviews and suggestions directly coming from real users to modify and improve the app. ICT design principles, indeed, support the idea of user involvement in the development of technology (Bailey et al. 2013b). Best practices emphasize the advantages of a staged iterative process that incorporates inclusion and feedback from key stakeholders (Fersini et al. 2010). This has two advantages: first, the inclusion of stakeholders allows taking advantage of user knowledge and suggestions; second, it expands prospects for stakeholder acceptance of technological change and increases ownership in and championing of the project's success (Bailey et al. 2013a; Lupo 2014).

This discourse suggests that if on the one hand a private user may be a fundamental protagonist of innovation, on the other hand this creative process cannot be executed without the cooperation of different actors, such as users, the ICT industry, and stakeholders. Indeed, an essential factor for the successful adoption of technology regards the relationship

between stakeholders' promotion to support the user-base consolidation. The concept of "critical mass" focuses on the number of users as a significant factor in ICT growth; as the number of users grows, the technology tends to gain momentum and starts growing through a so-called "self-reinforcing process" (Hanseth and Aanestad 2003). If a self-reinforcing process is not activated, positive returns in the terms of effectiveness and performance will not be generated and the app will never take off.

In the case of Anthea, the judiciary substantially supported the service by signing the Anthea Protocol (see Section 4.2). However, differently from the Collega experience, which is characterized by a large lawyers' adoption, in the case of Anthea, lawyers were mainly hostile and considered the system an obstacle to their business rather than an opportunity. Therefore, Anthea's developer opted for a different promotion strategy involving directly the public and particularly potential users as divorced couples and related associations. The creator started a very active promotion campaign with meetings, conferences, and book presentations on the topic, which promoted the app's diffusion. Thus, direct user support can be a formidable flywheel for ICT innovation and diffusion. However, without stakeholder mediation (lawyers in Anthea's case), it may entail unpredictable counter effects.

In the Anthea case, the involvement of inexperienced users as divorced couples and related associations brought new suggestions and requests and thus represented an incentive to expand the services offered by the app and the context of the app. Sometimes, users do not see the opportunities offered by a service but focus on what is not available or what they would like, as the creator said during an interview: *«It was like investing in the development of car accessories to sell it»*. Therefore, it was necessary to expand the offer to other services toward other social security sectors: personal and health care, fighting against gender-based violence and bullying, assistance for the elderly and disabled, and management of pets after divorce or death of owners.

Due to these circumstances, Anthea became a component of a widespread "Mai Soli Project" (in English *Never Alone Project*) and was supported and lobbied at the political level. The involvement of politics supported the draft of a law on separations and divorces involving the use of enabling technologies and an Italian region joined and financed the project with European Social Funds (ESF). These developments promoting the entire *Mai Soli Project* represented a further tremendous effort for Anthea's diffusion.

However, Anthea's inclusion in the larger project entailed a radical change in the original configuration in terms of the organizational, technical, and institutional components involved and, consequently, in the size of the project itself. This significantly increased the complexity. First, it became necessary to substitute the initial small ICT firm with a more prominent company that could involve 20 programmers to develop apps on each thematic area of the Mai Soli Project. Second, the new project required the inclusion of other partners involved in the telemedicine and security activities. Third, the redefinition of technology and the new project necessitated the recruitment and training of specific operators to deal with users for the services offered. It is plausible that integration problems could arise in the coordination with other organizations offering the same services (law enforcement, health service, associations, etc.).

Therefore, differently from what intuition would suggest, the widespread user adoption of an app may become its Achille's heel. The adopted solution that expanded the Anthea Project to meet the direct demands of users and politics incentivized the creation of a "critical mass" of users and the activation of the "self-reinforcing process" (Hanseth and Aanestad 2003). In particular, the several services (from divorce management to health services), the technical components (the more sophisticated technology adopted), the legislation (the rules that need to be changed and adopted), the organization (number and level of the subjects involved), and the institutional level (involvement of many sectors of the public administration) have been modified in such a way to increase the system's complexity dramatically, making it difficult to achieve a new alignment. The project's ex-

cessive complexity was one of the main reasons for the failure of e-justice projects in several national contexts (Carnevali 2010; Contini and Lanzara 2014; Lupo 2014; Carnevali 2009).

Projects that are developed from a bottom-up approach—such as COLLEGA and ANTHEA—are sensibly affected by growing size and complexity (Maeda 2006). The more the creators' manage to stay focused, the more they can have a chance of success; yet, the more they expand, the more they risk getting out of control and failing (Carnevali 2019, 2009). Several European experiences of e-justice system development provide examples of failures related to the lack of attention posed by developers to comply with the so-called "maximum manageable complexity" (Contini and Lanzara 2014). This refers on the one hand to how many entanglements can be embodied in the online procedure without turning them into a limitation for performativity and on the other hand to how many investments and maintenance costs are considered tolerable in terms of affordability and sustainability (Carnevali 2019). This is the reason why the experience of successful ICT and e-justice projects suggests the use of an incremental approach when developing systems. The case of COLLEGA's development confirms the efficacy of this approach given that it has been characterized by several incremental stages: first, the design and implementation of a simplified version of the app with less functionality than the actual version; second, the implementation of tests with real users; and third, the improvement of the app based on tests and suggestions, leading to the introduction of new functionalities.

### 5.2. Features of New Technology: Ubiquitous, Accessible, Modular, and Interoperable

Looking at the experience of web 2.0 apps and services, it is appropriate to question how it is possible for a user to ideate complex systems without an ICT background. One of the key elements of this phenomenon is the intrinsic nature of new technologies based on mobile phones and internet services. First, the apps and web services are based on ubiquitous and easily available new technologies. Ubiquitous technology describes the current environment of human–technology interactions in which the user operates several computing systems and equipment simultaneously in the course of normal activities and may not even be aware that these machines are performing their own tasks, actions, and operations (Nucera et al. 2018). This is particularly facilitated by the portability and accessibility of these technologies incorporated in machinery used every day, such as telephones, personal computers, cars, or home automation systems (Greenfield 2010). This aspect means that apps and web services can easily reach a large user base and diffuse in users' everyday activities and users may see the opportunities offered by new technologies for creating additional services and applications in different contexts.

The interview with COLLEGA'S creator acknowledged that several apps available at the time of COLLEGA's creation had inspired him. These apps allowed creating a match between a user in search of a specific service and another user acting as a provider of the service. In particular, Uber, an app offering services that include peer-to-peer ridesharing, ride service hailing, food delivery, and a micromobility system with electric bikes and scooters, has inspired COLLEGA's ideator. The apps inspiring COLLEGA's creation were born in the context of the diffusion of sharing economy initiatives that review the traditional economic model with a new one in which independent individuals rent or "share" things like their cars, homes, and personal time with others in a peer-to-peer fashion. The intuition of COLLEGA's creator was that legal activities could also be shared in a peer-to-peer manner with the support of a phone and an online app.

The choice of basing technological development and ideation on already existing models reflects the concept of "installed base" (Hanseth and Aanestad 2003; Hanseth and Lundberg 2001; Ciborra and Lanzara 1994; Kallinikos 2009; Lanzara 2009; Lupo 2014; Velicogna and Contini 2009) belonging to the ICT and e-justice literature. The concept of installed base refers to the technological solutions, institutional arrangements, organizational practices, and legal frameworks that are already established when a new system is developed (Kallinikos 2009). Hanseth and Lyytinen (Hanseth and Lyytinen 2016) suggested that designers may reduce adoption barriers and safeguard existing capabilities

by basing the implementation stage of an information system on an existing installed base (Hanseth and Lundberg 2001). Notably, relying on an installed base can also produce problems, for example, some installed base components resist change and may hinder the evolution of a technology. (Lanzara 2009) described the dual character of an installed base: it "constitutes a pool of available resources that can be turned into convertible and usable materials"; on the other hand, it can foster inertia and hinder "the development of new configurations" (Lanzara 2009, pp. 11, 22). In the case of COLLEGA, the installed base is constituted both by the existing apps of the sharing economy that inspired its ideation and by the technological components such as web server, geolocalization, and chat technology that are combined in the app.

A further characteristic of new technologies is the fact that they are modular and therefore can be combined through different solutions to propose and design different typologies of services similar to the cases of the two investigated systems. In the case of COLLEGA, the creator combined technologies and functionalities such as geolocalization, internal chat, and exchange of documents to provide an online service that supports the activities of a regular lawyer. In the case of ANTHEA, developers utilized a standard development platform based on the shared agenda technology, which was used and diffused in all organizations where it is necessary to manage activities requiring the intensive and coordinated participation of people not residing in the same place. Additionally, in ANTHEA, the shared agenda is integrated with a management system for exchanging and archiving documents, an e-form for making payments, and a chat module for rapid communication.

The combination of several components in a technology reflects a tenet of ICT design theory, which is modularity. (Hanseth and Lyytinen 2016; Lanzara 2009; Lupo 2014) have indicated that system development based on an infrastructure composed of different loosely coupled layers (or components) (Contini and Lanzara 2009), which are connected by gateways, can be essential to positive outcomes. Such an infrastructure fosters rapid evolution, and a change in one system component does not require the modification of the entire infrastructure. Moreover, the failure of a single component in a modularized architecture does not undermine the entire system (Velicogna and Contini 2009; Simon 1962; Contini and Lanzara 2009; Baldwin et al. 2000).

The history of the two systems, however, does not concern only the connection of technological elements but also the integration of technological, "human", and organizational components. The interconnection among such types of components, such as consumer technologies, developer involvement, user legitimation, and institutional commitment, reflects the concept of "assemblage" (Cooper 1998; Lanzara 2009). Based on the assemblage concept, performing technologies result from integrated and different loosely coupled components (Lanzara 2009)—organizational, technical, institutional, and regulative—connected to each other. However, the assemblage-type composition is not always an advantage. Every assemblage can change suddenly due to the changes affecting one or more components; in this case, it is necessary to find a new alignment to maintain the app's performativity (Contini and Lanzara 2014). As clarified in the previous section, this process particularly affected ANTHEA, which faced a misalignment of its components due to the increase in complexity with the inclusion in the Mai Soli Project (see Section 5.2).

*5.3. Negative Capabilities: Organizational Precondition for Technological Development by Users' Initiative*

The analysis described in the previous sections ascertained that users' initiative and the accessibility and availability of new technologies might become fundamental factors for the development of innovation, also in the justice sector. Moreover, the two examples confirm a precondition for innovation that is in contrast with the smart city literature, namely, the absence (instead of the active role) of public services.

The introduction of any type of innovation (including technological) in the absence of a public service is well described by the concept of "negative capability" (Lanzara 1993) described by Lanzara in his 1993 book. As presented in Section 3, negative capability refers

to the ability of private citizens to adapt and create useful practices, services, and routines in hostile environments, above all affected by a void of usual frameworks, infrastructure, and services provided by government and public administration. In this context, private initiative is more free and adaptive compared with government initiative that has to comply with the mechanisms and rules of complex bureaucracy. Clearly, to propose innovation in critical conditions, private initiative uses its available resources as human capital or economic resources. In the two case studies, human capital, particularly the knowledge of legal practice, is fundamental to activate "negative capabilities" and develop "expert-driven" (Prior et al. 2011; Potts et al. 2010) IT innovation.

Based on previous argumentation, COLLEGA's and ANTHEA's experiences are in contrast with the triple- and quadruple-helix model of the smart city literature that focuses on the engagement of government (in addition to universities, industries, and communities) for creating a productive infrastructure for the promotion of bottom-up technological ideas (Leydesdorff and Deakin 2011; Nam and Pardo 2011). The development of the two apps required the intervention of industry (the ICT company that practically realized the technology) and human capital skills, such as the knowledge of law and practices related to lawyers' activities (a knowledge typically spread through university institutions) and found inspiration in the absence of the government services given that the two apps filled the void with a service not provided by public institutions. Additionally, the interaction with official institutions has represented a barrier rather than a source of opportunities for the apps' creation.

The development of COLLEGA has inspired from the creator's experience in the context of routine operations of legal practices with reference to the searching and finding a substitute lawyer for the hearing or a colleague to perform any type of administrative activity in one of the Italian judicial offices. The Italian code of civil procedure provides the possibility for any lawyer in Italy who cannot attend a hearing to appoint a procedural substitute who will perform the functions of the appointing lawyer.[11] COLLEGA's creator was stunned with the time lost in finding a colleague who can replace him in court. This activity involved looking for a lawyer who is authorized to operate in the district and appeal or cassation court[12] getting in touch with him or her or their secretary, asking for availability to replace him in a hearing, describing the nature of the case, agreeing on a rate, and sending the documents related to the case. All these actions had to be repeated if a lawyer is not available for the substitution and each time a substitution is needed, thus resulting in a considerable loss of time for lawyers. The app's creator, with his idea, wanted to provide a more practical and faster system that would allow a lawyer to manage effectively the search for a replacement.

In the case of ANTHEA, the creator/user filled a void in the divorce proceeding services, activating negative capabilities in a hostile environment in which expected resources were missing. In the Italian justice system, divorce is considered only in legal terms, ignoring the significant interpersonal relationships between parties.[13] Parties' communication is also conceived only in a legal mode: the information exchanged, requests, or decisions have a legal effect. The medium is paper or electronic registered letter (notification), which is too rigid and formal, not allowing for rapid and direct negotiation. The registered letters are official acts and can be utilized by a party as evidence in the dispute.[14] Therefore, they can upset interlocutors, generate suspicions, and put them in a condition to take full advantage in the case. Even emails that ex-spouses often use for daily operational communications are cumbersome, not very immediate, and easy to cause miscommunications. Mobile chats

---

[11]   Law n. 247 of 31 December 2012 and article 102 of the Italian Code of Criminal Procedure.

[12]   Italian lawyers registered to the national bar associations can defend before any district court in the Italian territory. However, lawyers, in order to defend before some superior courts (Constitutional Court, Supreme Court of Cassation, Council of State of the Italian Republic, Superior Court of Public Waters) have to be registered in the list of lawyers authorized to practice before higher jurisdictions (Law 247 of 31 December 2012).

[13]   Ibidem note 9.

[14]   Art. 7 of the law 20 November 1986 n. 890 on the notification by post disciplines on the use of registered letters as communication means with legal value.

and phone calls suffer even more from the psychological and emotional loads, thus they frequently lead to out-of-control discussions.

The court's legal approach does not move in the direction of "demilitarizing", stabilizing, and improving ex-spouses' relationships by basing them on calm and civil tones. Additionally, with a significant focus on legal aspects, the Italian justice system did not invest in a service that may allow an orderly daily communication between parties for the organization of commitments (such as the management of children), the making of payments, the exchange of documents, or sharing events. In the context of such a public service void and a real and practical need, the creator took action directly using the smartphone technologies available for apps (existing at the time of development) to develop a system that facilitates concrete and correct socio-economic management of the relationship between divorced couples.

The two case studies demonstrate that private initiative is able to adapt and create useful practices and services for filling the void of public institution services. The concept of negative capability is reflected in the ability of COLLEGA's and ANTHEA's creators in ideating a technological solution under critical circumstances. Based on this, "negative capability" refers to the capacity of private initiative in proposing innovation in not only the absence of public services but also when the role of institutions represents an obstacle for development. This is, for instance, the case for COLLEGA. The interview acknowledged that COLLEGA's creator looked for the support of the national bar association to realize his idea. In particular, the bar association, by making the data on registered lawyers available, would have made it possible a more reliable method of identification. The bar association did not provide its support, because at the time of COLLEGA's development the association was already planning the implementation of a similar application that successively did not achieve the desired development and dissemination objectives. Due to the missed collaboration with the bar association, COLLEGA's method of identification does not automatically check whether the lawyer using the app is registered to a bar association or not (in Italy, only lawyers registered to a local bar association can represent a client in a court)[15] In a context where the institutional role is against innovation, negative capability describes the capacity to activate new resources and ideas and travel new routes when the traditional ones are inaccessible. To circumvent the unavailability of the national bar association data, the developer searched for new collaborations activating a partnership with the AIGA, which provided special support to the technology and the dissemination of information on the app. As described previously, users who are part of the AIGA benefit from the section reserved for them in the application; AIGA users register via the link contained in the email sent by the association and searches can be restricted to the AIGA members.

Differently from COLLEGA, in the case of ANTHEA, the role of public institution and particularly of politics provided a considerable boost to the service with its inclusion in the Mai Soli Project. Thanks to lobbying actions initiated toward members of parliament, a draft law on separations and divorces involving the use of supporting technological systems was drafted. Then, lobbying pressure ran an Italian region to join and finance the project with European Social Funds (ESF) for an amount of EUR 4.5 million in 3 years. Political support gave visibility and concreteness to the entire Mai Soli Project and represented a further tremendous incentive for ANTHEA's diffusion. However, as mentioned above, ANTHEA's inclusion in the large project entailed a radical change in the original configuration of the "assemblage" in terms of the organizational, technical, and institutional components involved and consequently in the size of the project itself, thus bringing a significant increase in complexity and consequential chances of failure.

The examples clarify that even if technological innovation may arise in the context of the absence of public service, the role of public institutions is still important because it may affect, for better or worse, the evolution of the innovation introduced. In the particular case

---

15 Article 81 of the Italian Civil Procedure Code.

of e-justice, the role of public institutions is even more significant given that services require legitimation from the parties involved and legalization from the public institutions and judicial system to operate with legal performativity (Mohr and Contini 2014). In the case of ANTHEA, legitimation relies on the voluntary and explicit adhesion of ex-spouses who accept the app as a "mediation technology" (Thompson 1967) in their new relationship and thus authorize institutions—social services and the judge in charge of the case—to interact and share the trends of family management. In addition, to ensure ANTHEA's legalization and allowing its usability in court, a soft law—called the "ANTHEA Protocol"—was drafted. The protocol disciplines the use of the app in the context of civil proceedings in the court that intend to adopt it. To date, three Italian county courts signed the protocol. Moreover, an Italian district court's positive evaluation certified the product's validity, thus supporting its legitimation and legalization in divorce procedures.

Based on the previous argumentation, the analysis is not completely in contrast with the quadruple- and triple-helix theories; however, it better clarifies the ambiguous but necessary role of public institutions in different contexts and in the different phases of technological innovation introduction.

## 6. Conclusions

The analysis of the two smart apps developed by the two Italian lawyers presented here, allowed putting in evidence some notable patterns. Both technologies have been ideated by the two lawyers who, aside for being passionate about new and smart technologies, do not have an ICT background. As already mentioned, ubiquitous, easily available, and modular technologies allow a "smart" creator, even if not supported by a technical ICT expertise, to assemble an application around a specific task by connecting different modules, such as GPS or chat technologies. We have highlighted in this paper how the assembly of different components reflects the different principles of the literature on ICT design and development, such as "installed base use" (Ciborra and Lanzara 1994; Hanseth and Lundberg 2001; Hanseth and Lyytinen 2016; Kallinikos 2009; Lanzara 2009; Lupo 2014; Velicogna and Contini 2009) and the "assemblage of different loosely coupled components" (Contini and Lanzara 2009). On a global scale, these processes are common to the development of apps in the sharing-economy market from which the two apps were inspired.

The analysis of the two applications acknowledges that the involvement of the creator/user in the development of the app leads to some advantages. First, despite the lack of ICT background, the creator/user, in the present cases a lawyer, contributes to the project with their knowledge of the legal background, essential for the conception of the project idea and for its success. The legal expertise contributing to the project implementation acknowledges the importance of the role of human capital and knowledge spread by university (one of the components of the triple-helix model) for technological innovation. Second, the creator/user needs to involve ICT technicians for the effective development of the technology, as happened in both case studies, creating the basis for cross influence between different backgrounds and allowing the creation of multi-disciplinary teams. This cross fertilization suggests the extension of some professional training, such as law and medicine, by adding ICT topics and app creation. Third, both the cases acknowledge that the implementation of an app by one of its main users brings the advantage that the developer can act as a user and directly test the system in the first place. This facilitates the implementation of an incremental approach to development through an iterative process that incorporates feedbacks from key stakeholders (Lanzara 1993; Ciborra and Lanzara 2017).

In addition to these aspects, the analysis allowed comparing the triple- and quadruple-helix models with the reality of smart apps for lawyers created by creators/users. In both cases, the role of the private sector, government, citizens, and universities is important for creating the basis of app development. However, and in addition to smart cities tenets previously mentioned, the two case studies put in evidence the propulsive thrust of the lack of an efficient public service for the design of smart apps such as COLLEGA and ANTHEA. The absence of a public service and the lack of institutional support represent the adverse

conditions under which negative capabilities allow designing and developing a smart idea, such as the two apps analyzed. In the context in which public services and support are absent, the private initiative is capable of taking the advantage of negative capabilities to design and develop the missing service. This is also due to the fact that private initiative compared with government action is more free and adaptive given as it does not (or does to a lesser extent) have to comply to the mechanisms and rules of complex bureaucracy (Ciborra and Lanzara 2017; Contini and Lanzara 2014).

Finally, the analysis has also shown that although the initial absence of public services underlies the creation of online services created by lawyers, their evolution and survival often cannot be guaranteed without subsequent public institution involvement. On the one hand, technologies in the legal environment in order to have legal performativity (Nam and Pardo 2011; Maeda 2006) need legalization offered by the judicial system (see for instance the Anthea *protocol* in the Anthea case study). On the other hand, public institutions can ensure the longevity of the project through new investments. The relative high complexities that derive from this, in some cases (as the Anthea case), are potential countereffects that may hinder the successful deployment of the service.

**Author Contributions:** Conceptualization, methodology and theoretical framework: G.L. and D.C.; investigation, analysis, and draft of Collega's sections: G.L.; investigation, analysis and draft of Anthea's sections: D.C.; writing—original draft preparation, review and editing: G.L. and D.C. Both authors are contributors to the article statements. All authors have read and agreed to the published version of the manuscript.

**Funding:** This research was founded by internal IGSG-CNR Project RIGIUSS-Next—Riforme della giustizia per lo sviluppo delle società contemporanee, programma 'Next Generation EU' e PNRR.

**Institutional Review Board Statement:** The study was conducted according to the guidelines of the CNR (Consiglio Nazionale delle Ricerche) ethical committee. Given that the qualitative study involved a limited number of observations and that data gathering operations did not imply risks for data protection, human dignity and health and bioethics, the ethical approval was not necessary.

**Informed Consent Statement:** Informed consent was obtained from all subjects involved in the study without a specific form (via mail and during a public workshop). Proof of consent is available from the editor.

**Data Availability Statement:** This study is based on qualitative data gathering through semi-structured interviews. Data are described within the paper and not available in original form (as recordings) for privacy reasons.

**Conflicts of Interest:** The authors declare no conflict of interest.

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
