# Peer review of "Smart Justice in Italy: Cases of Apps Created by Lawyers for Lawyers and Beyond"

_laws, 2022_

Round 1

Reviewer 1 Report

I like the article and learned from it. It is both theoretically interesting and of practical interest. The case studies are nicely done and address the theoretical ideas. And readers will also get practical insights on how someone might develop an app that works for the legal profession. The writing is pretty good. There are a few odd uses of language for a US reader, such as using the word humus where it would be common to use the word soil. My only caveat is about my own expertise. I do not know the literature on apps and how they are developed and succeed or not. I do think the article has merit for a generalist reader with interest in the legal profession, which is how I would define myself. I recommend publication as you can see from my evaluations. I enjoyed reading this. 

Author Response

Dear Reviewer 1,

Thank you for your review and for your interest in our work. You only indicated some changes relative to the US speaking language. We think it is better to stick to the UK English style, given that the paper passed a professional language revision with a UK style. 

Thank you and best regards.

The authors

Reviewer 2 Report

I wonder how the focus on two (!) Italian apps can contribute to the international (!) debate on smart justice. The submitted article might be of limited overall value in this respect.

I opine that the key issue is the (too) narrow focus on two Italian apps. From a global perspective it would be of greater interest to read a broader study / commentary which includes comparable solutions from various jurisdictions. If that is unfeasible to implement, I would suggest to make the contribution more specific and explicitly limit it to Italy. In this case, the reader want to be introduced to more detailed Italian rules.

Author Response

Dear Reviewer, 

thank you for your review and for your interest in our work. In your review, you questioned the limited value of the article due to the fact that the scope is narrowed to two Italian cases.

With reference to this comment, we clearly emphasized in the methodology section that the analysis is based on a case-study method, therefore based on a qualitative (of course intrinsically not statistically relevant) and in-depth analysis. Additionally, we support the idea that selecting the case studies in the Italian legal context gives us easier access to data, allowing us to operationalize the in-depth analysis. Additionally, the Italian legal context is consistent with other civil and even common law countries because inserted in the sphere of influence of international organizations driving the harmonization of law as European Union, Council of Europe, and United Nations. Also for this, cases of children management in divorce cases and lawyers’ substitution in proceedings are common legal subjects in almost all national contexts.

However, in order to make the contribution more contextualized, as you suggested, we specified the previous statements in the paper with two new sentences, and we added a set of notes referring to the Italian legal framework (notes 51, 52, 56, 76, 77, 79).

Best Regards

The Authors

Round 2

Reviewer 2 Report

I strongly suggest to change the title of the contibution if the focus is limited to the two Italian apps. The authors' arguments on why they see the contribution representative for the EU as a whole are not convincing in my opinion. The paper itself is worth publishing though - if the title reflects the narrow focus on Italy. Otherwise the title is misleading.

Author Response

Dear Reviewer,

we followed your suggestion and we modified the title accordingly. Now the scope is more narrow.

Best Regards

Giampiero Lupo